# EXPLORE-INSTRUCT: Enhancing Domain-Specific Instruction Coverage through Active Exploration

**Fanqi Wan**[1][*] **Xinting Huang**[2][†] **Tao Yang**[1]**, Xiaojun Quan**[1][†]**, Wei Bi**[2]**, Shuming Shi**[2]

[1]School of Computer Science and Engineering, Sun Yat-sen University, China
[2]Tencent AI Lab
{wanfq, yangt225}@mail2.sysu.edu.cn, quanxj3@mail.sysu.edu.cn,
{timxthuang, victoriabi, shumingshi}@tencent.com

## Abstract

Instruction-tuning can be substantially optimized through enhanced diversity, resulting in models capable of handling a broader spectrum of tasks. However, existing data employed for such tuning often exhibit an inadequate coverage of individual domains, limiting the scope for nuanced comprehension and interactions within these areas. To address this deficiency, we propose EXPLORE-INSTRUCT, a novel approach to enhance the data coverage to be used in domain-specific instruction-tuning through active exploration via Large Language Models (LLMs). Built upon representative domain use cases, EXPLORE-INSTRUCT explores a multitude of variations or possibilities by implementing a search algorithm to obtain diversified and domain-focused instruction-tuning data. Our data-centric analysis validates the effectiveness of this proposed approach in improving domain-specific instruction coverage. Moreover, our model's performance demonstrates considerable advancements over multiple baselines, including those utilizing domain-specific data enhancement. Our findings offer a promising opportunity to improve instruction coverage, especially in domain-specific contexts, thereby advancing the development of adaptable language models. Our code, model weights, and data are public at https://github.com/fanqiwan/Explore-Instruct.

## 1 Introduction

Large language models (LLMs) have proven to be exceptional in following a variety of instructions. Recent studies have shown that LLMs of a smaller size may address a variety of circumstances and adapt to various requirements if fine-tuned with a small number of instruction-following demonstrations (Zhou et al., 2023). An important challenge thus is to construct diverse instruction-tuning data (Gudibande et al., 2023).

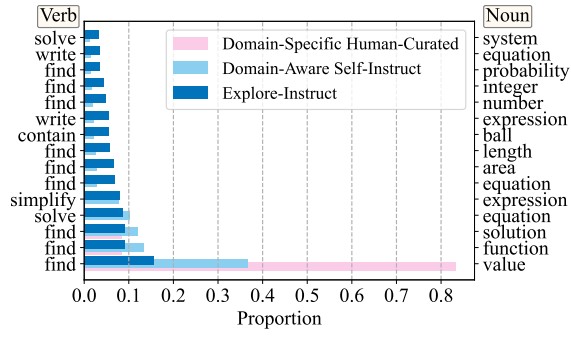

Figure 1: The top-15 most common verb-noun pairs in instructions of previous methods and EXPLORE-INSTRUCT for math problem-solving. It reveals an over-concentrated range of instructions in prior methods, while EXPLORE-INSTRUCT offers broader coverage.

Early research utilized human-curated instruction-tuning data covering a wide array of tasks, ranging from FLAN (Chung et al., 2022) to SUPER-NATURALINSTRUCTION (Wang et al., 2022b). More recently, SELF-INSTRUCT (Wang et al., 2022a) approach has been proposed to amplify the diversity of instruction-tuning data by querying instruction-tuned LLMs (Xu et al., 2023b; Ji et al., 2023). These advancements signify major progress in the evolution of LLMs that can cater to a diverse set of user needs.

In addition to those excelling in general domains, there is a growing demand for LLMs proficient in specific domains[1], which can be naturally obtained by domain-specific instruction-tuning. Like their counterparts on general domains, the creation of domain-specific instruction-tuning data can be split into two categories: human-curated and data generated by LLMs. Specifically, the re-utilization of human-curated data examples has been explored in several scenarios, e.g., machine translation, arithmetic tasks (Liu and Low, 2023; Zhang et al., 2023;

---

[*] Work was done during the internship at Tencent AI lab.
[†] Corresponding authors.

[1]In this context, 'domain' refers to specific use cases, such as writing assistance, mathematical problem-solving, or coding assistance. Each of these presents its own unique set of challenges, methodologies, and requirements.

Jiao et al., 2023; Wang et al., 2023), while the development of domain-aware self-instruct examples is a research area of growing interest (Chaudhary, 2023; Luo et al., 2023). However, current methodologies for collating domain-specific instruction-tuning data fail to encapsulate the wide range of potential instructions within a given domain as supported by the results depicted in Figure 1. This is primarily due to their over-reliance on human curation, which tends to be biased towards popular NLP tasks and lacks diversity. Consequently, these methods fail to encompass the true diversity of tasks in the domain (Wang et al., 2022a).

To enhance the coverage of domain-specific instruction-tuning data, we introduce a novel instruction paradigm named EXPLORE-INSTRUCT. We posit that the domain space is inherently structured akin to a tree, reminiscent of cognitive science ontologies (Newell et al., 1959). Drawing from the essence of classical search algorithms and incorporating the power of LLMs, EXPLORE-INSTRUCT is conceived to actively traverse the domain space and generate instruction-tuning data, *not* necessitating a predefined tree structure. Specifically, our approach employs two strategic operations: lookahead and backtracking explorations. In this context, lookahead delves into a multitude of potential fine-grained sub-tasks, thereby mapping out a complex network of tasks, whereas backtracking exploration seeks alternative branches to widen the search boundary, hence extending the domain spectrum. Through empirical analysis, we observe that a simple Depth-First Search (DFS) implementation proves to be sufficient in navigating vast domain spaces with minimal overhead.

To validate the efficacy of EXPLORE-INSTRUCT, we select three disparate domains - *rewriting*, *brainstorming*, and *math* [2] - as our experimental testbeds. Each of these domains encapsulates a unique use case and demands different skill sets, underscoring the versatility of our proposed approach. After applying EXPLORE-INSTRUCT, we analyze the data obtained from each domain and detect a comprehensive level of coverage. Our findings suggest that the LLMs can explore not only a wide breadth of tasks within a specific domain but also delve deeply into these tasks and decompose them

into fine-grained sub-tasks. Fine-tuning domain-specific models using data garnered from the proposed method led to outperforming the baselines in both automatic evaluation and human evaluation.

Our contributions are summarized as follows:

- We highlight the limitations in current domain-specific instruction-tuning data, emphasizing the inadequate coverage within individual domains.

- We introduce EXPLORE-INSTRUCT, a novel strategy for enhancing domain-specific coverage, which employs an active exploration method using LLMs.

- We empirically confirm the efficacy of EXPLORE-INSTRUCT through both data-centric analysis and model performances, showcasing its substantial improvements in instruction coverage and superior performances over multiple baselines.

## 2 Related Work

### 2.1 Instruction-Tuning for General Domains

Recently, there has been a growing interest among NLP researchers in the area of instruction-tuning. Early research such as FLAN (Wei et al., 2022) and T0 (Sanh et al., 2022) utilize human-curated open-source datasets that encompass a wide range of tasks. Building on this foundation, subsequent efforts such as SUPER-NATURALINSTRUCTION (Wang et al., 2022b), MetaICL (Min et al., 2022), and FLAN-T5 (Chung et al., 2022), seek to expand the scope and quality of the collected datasets by combining more tasks, introducing in-context examples and chain-of-thought data. More recently, to reduce the cost of manual involvement, SELF-INSTRUCT (Wang et al., 2022a) is proposed to produce data via instruction-tuned LLMs, such as ChatGPT (OpenAI, 2023a) and GPT-4 (OpenAI, 2023b). For example, Alpaca (Taori et al., 2023) uses ChatGPT to generate 52k instruction-tuning data from a small manually written instruction seed set, LaMini-LM (Wu et al., 2023) develops a dataset of 2.58M examples based on both existing and generated instructions. Dynosaur (Yin et al., 2023) uses the metadata of datasets to generate instruction templates and determine the data fields with ChatGPT. Baize (Xu et al., 2023b) constructs multi-turn instruction-tuning data by requesting

---

[2] The terms *rewriting* and *brainstorming* are drawn from existing works (Ji et al., 2023). The rewriting domain covers the tasks of refining and modifying text, whereas the brainstorming domain is oriented toward generating novel and open-ended passages.

ChatGPT to engage in a conversation based on a given topic. Dromedary (Sun et al., 2023) and WizardLM (Xu et al., 2023a) enhance the alignment and complexity of generated instructions respectively by heuristically modifying the input prompts.

## 2.2 Instruction-Tuning for Specific Domains

In addition to the works focusing on general domains, there is a growing demand for LLMs experienced in specific domains. Following the success in general domains, the re-utilization of human-curated public datasets has been employed in several scenarios, such as arithmetic (Liu and Low, 2023), writing (Zhang et al., 2023), machine translation (Jiao et al., 2023), and information extraction (Wang et al., 2023). Another line of work adopts the self-instruct in specific domains. For example, CodeAlpaca (Chaudhary, 2023) and Wizard-Coder (Luo et al., 2023) develop the domain-aware self-instruct for the coding tasks. Our proposed EXPLORE-INSTRUCT differs from these methods by actively traversing the domain space with lookahead and backtracking explorations to enhance the instruction coverage.

## 3 Methods

In this section, we begin by introducing the domain space representation in Section 3.1. Subsequently, we present the active exploration strategy, which encompasses both lookahead and backtracking explorations, in Section 3.2. Finally, in Section 3.3, we provide a detailed description of the EXPLORE-INSTRUCT implementation.

### 3.1 Domain Space Representation

We posit that the coverage of domain-specific instruction is influenced by two key factors: *breadth*, referring to the inclusiveness of different task categories within the domain, and *depth*, which relates to fine-grained task decomposition. The critical role of breadth is highlighted by its capacity to promote understanding of various categories of tasks within a domain (Hendrycks et al., 2021; Huang et al., 2023). On the other hand, depth promotes thorough understanding and precise problem-solving as underscored by task decomposition studies (Huang and Chang, 2022; Khot et al., 2023). Therefore, we model the domain space as a tree structure $\mathcal{T}$, with nodes $V$ representing tasks and edges $E$ denoting the hierarchical relationship between a task and its decomposed sub-tasks. In this way, the domain space representation offers a structured approach to obtain comprehensive domain-specific instruction-tuning data.

## 3.2 Active Exploration Strategy

The proposed EXPLORE-INSTRUCT comprises two fundamental operations: (1) lookahead exploration and (2) backtracking exploration, as illustrated in Figure 2.

**Lookahead Exploration**  The first operation involves exploring the domain space in the depth direction, thereby mapping out a complex network of potential fine-grained sub-tasks. Specifically, given a task $V_i$, the lookahead exploration utilizes a LLM to decompose $V_i$ into $M$ distinct sub-tasks that differ from those already present in $\mathcal{T}$. An example of the prompt template for lookahead exploration is as follows[3]:

> You are asked to propose some new sub-tasks for the target task given the current exploration state.
>
> Here are the requirements:
> 1. The skills required to perform a sub-task belong to the skills required to perform the target task, and the former is a subset of the latter.
> ......
>
> Current exploration state: {Exploration State}
> Target task: {Target task}
> Generate $M$ new sub-tasks with the corresponding reasons:

Here, the placeholder {Target task} refers to the specific task $V_i$ to be decomposed. The placeholder {Exploration State} is composed to approximately represent the existing progress in $\mathcal{T}$, which directs the exploration towards regions that are yet to be thoroughly examined.

**Backtracking Exploration**  In addition to lookahead exploration, backtracking exploration is another key operation in EXPLORE-INSTRUCT. It seeks alternative branches in the domain tree $\mathcal{T}$ to expand the search boundary and increase the diversity of tasks in the domain. Specifically, given a task $V_j$, we first perform the backtracking operation to find its parent task $V_i$ and then utilize a LLM to explore $M$ new sub-tasks of $V_i$ breadth-wise. The prompt used for backtracking exploration is similar to that employed for lookahead exploration.

### 3.3 EXPLORE-INSTRUCT Implementation

The detailed description of EXPLORE-INSTRUCT implementation can be broken down into two dis-

---

[3]Please refer to Appendix A for the complete prompt.

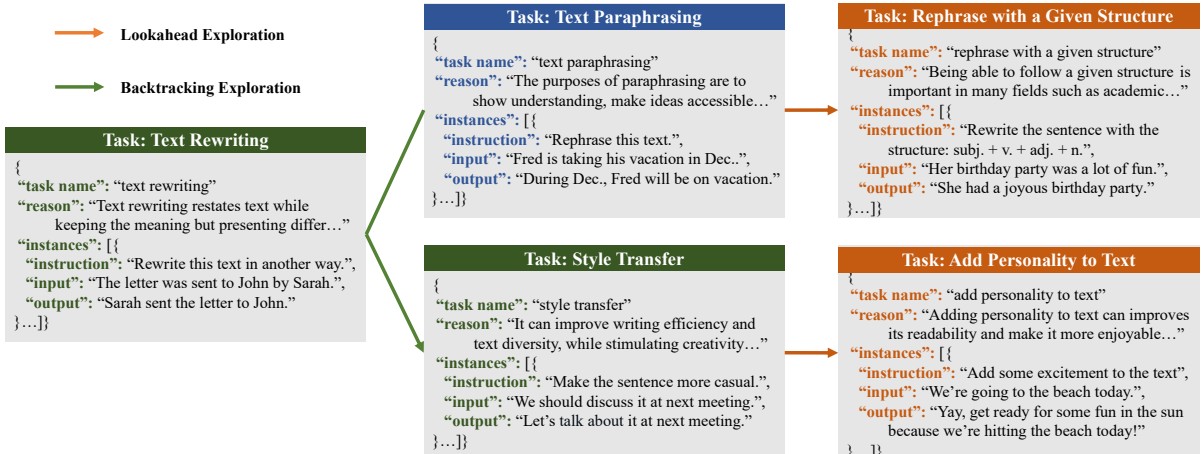

Figure 2: The overview of our proposed EXPLORE-INSTRUCT. It involves two strategic operations: (1) lookahead exploration, and (2) backtracking exploration. The lookahead exploration delves into a multitude of potential fine-grained sub-tasks, whereas the backtracking exploration seeks alternative branches to widen the search boundary.

tinct processes: (1) domain exploration, and (2) subsequent instruction-tuning data generation.

**Domain Exploration Strategy**   The exploration process begins at the root task of the domain and proceeds by traversing each node in a Depth First Search (DFS) manner, accompanied by either lookahead exploration or backtracking exploration. The stopping criterion for either action is determined by a maximum exploration depth $K$, or breadth $B$. By incorporating both lookahead and backtracking explorations, the process achieves effective exploration of the domain space, thereby enhancing the model's adaptability to a diverse range of user requirements.

**Instruction-Tuning Data Generation**   For each task in the domain tree, we follow prior work (Taori et al., 2023) and use a LLM to generate $N$ instructions and corresponding responses[4]. To ensure the diversity of instructions, we adopt a diversity filter during domain exploration and instruction-tuning data generation. Specifically, a decomposed sub-task or generated instruction is retained only if its ROUGE-L overlap with any existing task or instruction is less than a threshold.

## 4   Data-Centric Analysis

To illustrate the efficacy of EXPLORE-INSTRUCT from a data-driven perspective, we perform a data-centric analysis. We first introduce the baseline methods for generating instruction-tuning data that

we select for comparison. Subsequently, we perform a comprehensive statistical analysis of the data obtained for each domain and examine the domain coverage yielded by different methods.

### 4.1   Baselines

We select the following baseline methods for comparison with EXPLORE-INSTRUCT[5]:

**Domain-Specific Human-Curated**   This method collates data related to a specific domain from various open-source datasets. To ensure data quality, we carefully select datasets within the SUPER-NATURALINSTRUCTION and randomly sample 10,000 training instances[6].

**Domain-Aware Self-Instruct**   This method obtains domain-specific instruction-tuning data through a LLM. It is a simplified version of EXPLORE-INSTRUCT, with a maximum exploration depth of $K = 0$. Specifically, an initial domain-specific seed collection with diverse instructions is used. During each iteration, two examples are randomly selected from the seed collection. Subsequently, the newly generated instruction data is integrated into the seed collection, and the process is reiterated. To ensure a fair comparison, we randomly sample 10,000 training instances.

### 4.2   Data Statistics

Table 1 presents the essential statistics of the domain-specific instruction-tuning data generated using various methods, including Domain-Specific

---

[4]Please refer to Appendix A to find the prompt for instruction-tuning data generation.

[5]The implementation details of EXPLORE-INSTRUCT can be found in Section 5.3.

[6]The selected datasets can be found in Appendix C.

| Data Stat. | Brainstorming | Rewriting | Math |
|---|---|---|---|
| *Domain-Specific Human-Curated* | | | |
| Unique V-N pairs ↑ | 2 | 8 | 3 |
| Occur. of V-N pairs (Avg.) ↓ | 5000 | 778 | 1355 |
| Occur. of V-N pairs (Std.) ↓ | 1447 | 835 | 779 |
| *Domain-Aware Self-Instruct* | | | |
| Unique V-N pairs ↑ | 781 | 1715 | 451 |
| Occur. of V-N pairs (Avg.)↓ | 4 | 5 | 13 |
| Occur. of V-N pairs (Std.) ↓ | 14 | 14 | 68 |
| EXPLORE-INSTRUCT | | | |
| Unique V-N pairs ↑ | 790 | 2015 | 917 |
| Occur. of V-N pairs (Avg.) ↓ | 3 | 4 | 7 |
| Occur. of V-N pairs (Std.) ↓ | 13 | 12 | 24 |

Table 1: Statistics of verb-noun (V-N) pairs in the instructions obtained from Domain-Specific Human-Curated, Domain-Aware Self-Instruct, and EXPLORE-INSTRUCT in different domains.

Human-Curation, Domain-Aware Self-Instruct, and EXPLORE-INSTRUCT. Notably, the number of unique verb-noun pairs in the instructions produced by EXPLORE-INSTRUCT is higher than that of Domain-Specific Human-Curation and Domain-Aware Self-Instruct. Moreover, the average and standard deviation of the number of occurrences is lower. This phenomenon is apparent in the rewriting and math domains, whereas it is less pronounced in the brainstorming domain. To provide a more explicit demonstration, we compare the verb-noun pairs in the generated instructions whose frequency is above 10 between Domain-Aware Self-Instruct and EXPLORE-INSTRUCT in the math domain and show the results in Figure 3. The verb-noun pairs are distributed more uniformly in the instructions generated by EXPLORE-INSTRUCT.

Furthermore, we analyze the distribution of the average ROUGE-L overlap between the generated instructions as well as the existing instructions of baseline methods and EXPLORE-INSTRUCT. Specifically, for each generated instruction, we compute its average ROUGE-L score with the existing instructions generated before it, and the result distributions are shown in Figure 4. Compared with Domain-Specific Human-Curation and Domain-Aware Self-Instruct, the average ROUGE-L scores for EXPLORE-INSTRUCT are concentrated in regions with relatively smaller values.

These findings suggest that, compared with baseline methods, which exhibit an inadequate coverage of generated instruction-tuning data, EXPLORE-INSTRUCT can enhance the diversity and domain coverage of the instruction-tuning data through depth-wise and breadth-wise active exploration of the domain space.

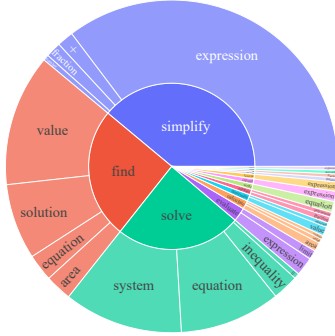

(a) Domain-Aware Self-Instruct

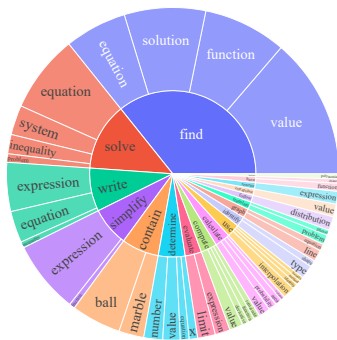

(b) EXPLORE-INSTRUCT

Figure 3: The root verb-noun pairs in instructions of (a) Domain-Aware Self-Instruct and (b) EXPLORE-INSTRUCT in the math domain, where the inner circle of the plot represents the root verb of the generated instructions, and the outer circle represents the direct nouns.

## 5 Experimental Settings

### 5.1 Benchmarks

To validate the efficacy of EXPLORE-INSTRUCT, we chose three distinct domains: *rewriting*, *brainstorming*, and *math*, as experimental testbeds. Each of these domains represents a unique use case and the scope of required skill sets varies differently, highlighting the versatility of our approach. The testbeds for the brainstorming and rewriting domains are derived from the corresponding categories in the translated test set of BELLE (Ji et al., 2023). While for the math domain, we randomly select 500 questions from the test set of MATH (Dan et al., 2021; Lightman et al., 2023). The statistics of these testbeds can be found in Appendix D.2.

Consistent with prior research (Peng et al., 2023; Chen et al., 2023), for the brainstorming and rewriting domains, we employ both automatic and human evaluations to evaluate the performance of domain-specific models. As for the math domain, we measure the performance using the Accuracy Rate metric in solving math problems.

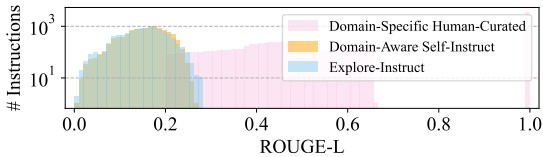

(a) Brainstorming

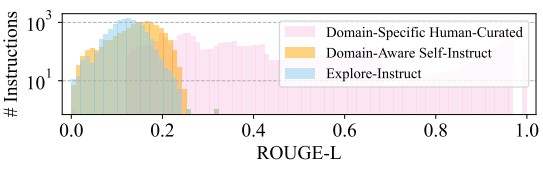

(b) Rewriting

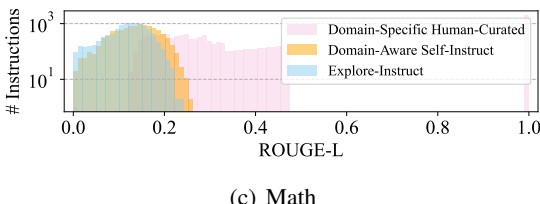

(c) Math

Figure 4: The distribution of average ROUGE-L overlap between generated and existing instructions of Domain-Specific Human-Curated, Domain-Aware Self-Instruct, and EXPLORE-INSTRUCT in different domains.

## 5.2 Explore-LM and Baseline Models

**Explore-LM** Explore-LM is a domain-specific assistant developed by implementing the proposed EXPLORE-INSTRUCT and fine-tuning the backbone model on the obtained instruction-tuning data.

**Baseline Models** To facilitate a meaningful comparison with our proposed Explore-LM, we adopt two baseline models, namely Domain-Curated-LM and Domain-Instruct-LM. These models are fine-tuned with the same backbone using instruction-tuning data generated from Domain-Specific Human-Curated and Domain-Aware Self-Instruct, as described in Section 4.1. Furthermore, we compare the performance of Explore-LM against ChatGPT, which serves as an upper bound in our evaluation.

## 5.3 Implementation Details

**EXPLORE-INSTRUCT** To construct the domain-specific instruction-tuning data, we actively traverse the domain space and generate data with a maximum exploration depth of $K = 2$. We set the maximum exploration breadth to $B = 8$ and $B = 6$ in the first and second depths, respectively. We set the number of exploration sub-tasks

to $M = 3$, and generate $N = 500$ instructions and responses. The filter threshold is set to 0.7. Once the process is complete, we randomly sample 10,000 training instances from each sub-task data to fine-tune the Explore-LM, with the number of instances in each sub-task serving as the sampling weights. To further enhance the performance of the domain-specific model, we increase the number of sampled instances to fine-tune the Explore-LM-Ext. We utilize gpt-3.5-turbo to perform domain exploration and data generation, with a maximum length of 4096, a temperature of 1.0, and a top-p of 1.0 during API requests. In contrast to Domain-Aware Self-Instruct, which performs the data generation process only, the primary additional cost of tokens associated with EXPLORE-INSTRUCT comes mainly from the domain exploration process. Since this exploration occurs at the task level and allows for simultaneous exploration of multiple tasks, the additional cost is relatively minimal and approximately less than 1%. Please refer to Appendix D.1 for the statistics of the final instruction-tuning data.

**Training Details** We utilize LLaMA 7B (Touvron et al., 2023) as the backbone model. We adopt AdamW optimizer with a learning rate of 2e-5 and the batch size is 128. The maximum length is 512. We train all models using 8 V100 GPUs with deepspeed Zero-3 offload for 3 epochs.

**Inference Details** We use a sampling method with a temperature of 0.7 to diversify outputs for the brainstorming and rewriting domains. For the math domain, we employ a beam search with a beam size of 10 to reduce output randomness and ensure more deterministic outputs. The maximum generation length during inference is 512.

## 6 Results and Analysis

In this section, we conduct both automatic and human evaluations to justify the validity of EXPLORE-INSTRUCT. Our Explore-LM fine-tuned with data generated from EXPLORE-INSTRUCT outperforms the baselines in both evaluations. Additionally, we perform in-depth analysis to investigate the influence of various factors, such as data structure, quantity, and quality on the performance of Explore-LM. Our analysis reveals that the maximum exploration depth has the most significant impact on performance, followed by the number of training instances and the utilization of the diversity filter.

| Automatic Comparison | Brainstorming | | Rewriting | |
|---|---|---|---|---|
| | Win:Tie:Lose | Beat Rate | Win:Tie:Lose | Beat Rate |
| Explore-LM vs Domain-Curated-LM | 194:1:13 | 93.72 | 50:38:6 | 89.29 |
| Explore-LM-Ext vs Domain-Curated-LM | 196:1:11 | **94.69** | 53:37:4 | **92.98** |
| Explore-LM vs Domain-Instruct-LM | 114:56:38 | 75.00 | 34:49:11 | 75.56 |
| Explore-LM-Ext vs Domain-Instruct-LM | 122:55:31 | **79.74** | 35:53:6 | **85.37** |
| Explore-LM vs ChatGPT | 52:71:85 | 37.96 | 11:59:24 | 31.43 |
| Explore-LM-Ext vs ChatGPT | 83:69:56 | **59.71** | 12:56:26 | **31.58** |

Table 2: Automatic evaluation results in the brainstorming and rewriting domains. It demonstrates that Explore-LM outperforms multiple baselines with a large Beat Rate and nearly matches the performance of ChatGPT. Additionally, with a substantial increase in training instances, the performance of Explore-LM-Ext can be further enhanced.

| Models | Math Accuracy Rate |
|---|---|
| Domain-Curated-LM | 3.4 |
| Domain-Instruct-LM | 4.0 |
| Explore-LM | 6.8 |
| Explore-LM-Ext | 8.4 |
| ChatGPT | **34.8** |

Table 3: Automatic evaluation results in the math domain. The results illustrate that Explore-LM achieves significant improvements on baseline models.

## 6.1 Automatic Evaluation

To evaluate the performance of Explore-LM in the brainstorming and rewriting domains, we follow Chen et al. (2023) to conduct an automatic evaluation with ChatGPT. We do not use `gpt-4` but use `gpt-3.5-turbo` as the evaluator since the limited quota of the OpenAI account. Specifically, we adopt pair-wise evaluation, where given a question and two responses from different models, we request ChatGPT to determine which response is better based on their helpfulness, relevance, accuracy, and level of detail, with the corresponding reasons and justifications[7]. To calculate the Beat Rate of a particular model, we divide the number of times the model wins by the sum of the number of times the model wins and loses. For the math domain, we utilize the automatic evaluation script from MATH to calculate the Accuracy Rate of model answers[8].

The automatic evaluation results are shown in Table 2 and Table 3. Explore-LM exhibits a significant performance advantage over multiple baseline models in all domains while utilizing an equal number of training instances. In the brainstorming and rewriting domains, Explore-LM achieves Beat Rates of 93.72 and 89.28, respectively, when

compared to Domain-Curated-LM. Similarly, when compared to Domain-Instruct-LM, Explore-LM achieves Beat Rates of 75.00 and 75.56 in the same domains. In the math domain, Explore-LM achieves an Accuracy Rate of 6.8, compared to Domain-Curated-LM's 3.4 and Domain-Instruct-LM's 4.0. Moreover, when we increase the number of instances, Explore-LM-Ext demonstrates even better performance, surpassing ChatGPT with a Beat Rate of 59.71 in the brainstorming domain.

## 6.2 Human Evaluation

To provide a more comprehensive and reliable evaluation, we incorporate human evaluation for both the brainstorming and rewriting domains. Specifically, we enlist three annotators to compare the answers generated by two models for the same question and indicate which model provided the better answer (win, tie, or lose). To maintain anonymity and prevent any potential bias, the evaluated models are kept anonymous, and the annotators remain unaware of which model generated each answer.

Figure 5 illustrates the results of the human evaluation. The comparison between the human and automatic evaluations demonstrates a general consistency, indicating that our approach is also qualitatively well-regarded by humans, underscoring its effectiveness. When comparing Explore-LM with ChatGPT, the latter generates answers that are preferred by humans in most cases. This preference can be attributed to the reinforcement learning with human feedback approach employed in ChatGPT.

## 6.3 Data Structure Analysis

To analyze the impact of data structure on the performance of Explore-LM, we conduct experiments in which we vary the maximum exploration depth within a range of 0 to 2 while sampling 10,000 training instances. We compare the performance of

---

[7]The evaluation prompt can be found in Appendix B.
[8]https://github.com/hendrycks/math

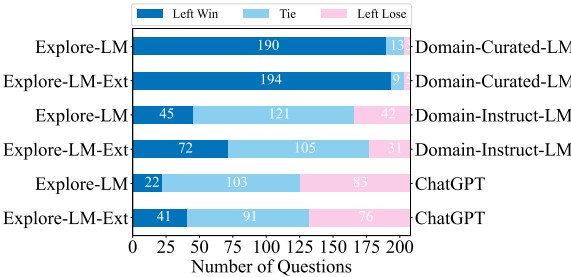
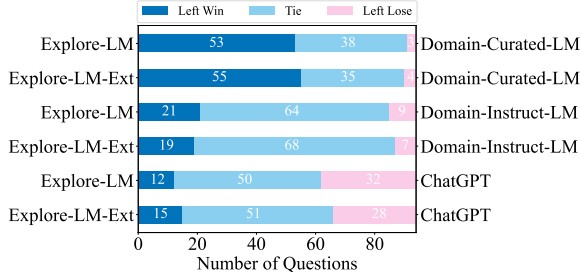

Figure 5: Human evaluation results in the brainstorming (Left) and rewriting (Right) domains.

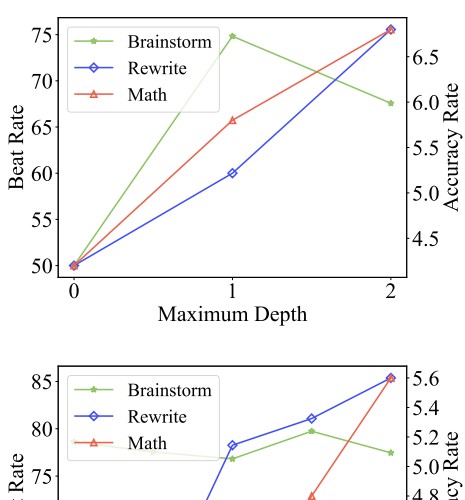
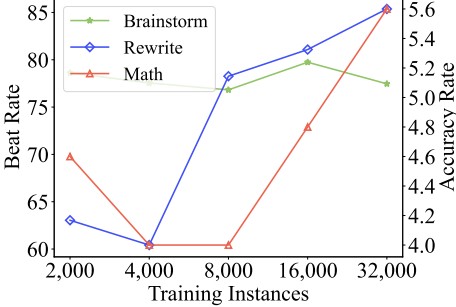

Figure 6: Performance of Explore-LM with different maximum exploration depths (Upper) and the number of training instances (Down).

| Models | Brainstorming Beat Rate | Rewriting Beat Rate | Math Accuracy Rate |
|---|---|---|---|
| Explore-LM | 75.00 (↑0.89%) | 75.56 (↑8.30%) | 6.8 (↑3.03%) |
| *w/o* filter | 74.34 | 69.77 | 6.6 |

Table 4: Results of the data quality analysis, where "filter" denotes the diversity filter which controls the diversity in the generated instruction-tuning data.

the maximum exploration depth can enhance the coverage of instruction-tuning data, thus improving the performance of Explore-LM across various domains and scenarios.

## 6.4 Data Quantity Analysis

We also investigate the influence of data quantity. To this end, we conduct a series of experiments in which we vary the number of training instances exponentially, ranging from 2,000 to 32,000 while keeping a consistent maximum exploration depth. We then compare the performance of Domain-Instruct-LM using 10,000 instances against that of Explore-LM using varying numbers of instances. The results are shown in Figure 6 (Down). Our findings indicate that Explore-LM using 2,000 instances outperforms Domain-Instruct-LM using 10,000 instances and even achieves a Beat Rate of nearly 80.00 in the brainstorming domain. Moreover, our results demonstrate that data quantity provides varying benefits across different domains. For instance, in the brainstorming domain, increasing the number of instances from 2,000 to 32,000 results in an increased performance of only 1.42%. In contrast, for the rewriting and math domains, this can lead to substantial performance gains of 35.42% and 21.74%, respectively.

## 6.5 Data Quality Analysis

In this section, we discuss the impact of data quality. Specifically, we conduct an automatic comparison between Domain-Instruct-LM and Explore-LM, both with and without the diversity filter men-

Domain-Instruct-LM and Explore-LM with varying maximum depth. The results presented in Figure 6 (Upper) illustrate that increasing the maximum exploration depth generally leads to significant improvements in model performance. Specifically, we observe an increase in performance of 51.52% and 61.90% for the rewriting and math domains, respectively. However, for the brainstorming domain, the model demonstrates optimal performance at a relatively small maximum depth. We posit that in domains with a comparatively small scope of search space, such as brainstorming, high coverage is already achieved at a shallow depth, and further increasing the exploration depth may result in duplicated tasks that reduce diversity, and ultimately negatively impact the model performance. Our findings indicate that increasing

tioned in Section 3.3. We maintain consistency in terms of the maximum exploration depth and the number of training instances. The results are shown in Table 4. It is observed that the performance of Explore-LM improves slightly across all domains with the use of the diversity filter. The brainstorming domain demonstrates a marginal improvement of only 0.89%, whereas the rewriting and math domains exhibit more significant improvements of 8.30% and 3.03%, respectively.

## 7 Conclusion

In this work, we introduce EXPLORE-INSTRUCT, a novel approach to enhancing domain-specific instruction coverage. Drawing inspiration from classical search algorithms, EXPLORE-INSTRUCT leverages the power of LLMs to actively explore the domain space and obtain diverse and domain-focused instruction-tuning data. Our experimental results demonstrate the efficacy of EXPLORE-INSTRUCT through data-centric analyses and model performance evaluations in the rewriting, brainstorming, and math domains, highlighting significant enhancements in instruction coverage and superior model performance compared to multiple baseline methods as demonstrated by both automatic and human evaluations.

## Acknowledgements

This work was supported by the National Natural Science Foundation of China (No. 62176270), the Guangdong Basic and Applied Basic Research Foundation (No. 2023A1515012832), and the Tencent AI Lab Rhino-Bird Focused Research Program.

## Limitations

The limitations of EXPLORE-INSTRUCT are discussed as follows. Firstly, our approach relies on the power of LLMs which have limitations in terms of their resource requirements and computational complexity, which may limit the scalability of our approach. Secondly, our study focuses on the enhancement of domain-specific instruction coverage and does not address other aspects of instruction-tuning, such as the generation of complex and challenging instructions or the identification and mitigation of toxic and harmful instructions. Future work is needed to explore the potential of our approach in these areas.

## Ethics Statement

We state that any research or application arising from this study is strictly authorized solely for research purposes. The BELLE and MATH test sets used in our work are obtained from public sources and do not contain any private information. Our research adhered strictly to the data usage policy.

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

## A  Prompts for EXPLORE-INSTRUCT

Here we provide the prompts for EXPLORE-INSTRUCT. The lookahead and backtracking exploration prompt is shown in Figure 7, and the domain-specific instruction-tuning data generation prompt is shown in Figure 8.

## B  Prompt for Automatic Evaluation

The prompt we use to conduct automatic pair-wise evaluation is shown in Figure 9.

## C  Datasets Collection of Domain-Specific Human-Curated

To obtain a collection of Domain-Specific Human-Curated datasets, we selected tasks related to each domain in the SUPER-NATURALINSTRUCTION and combined their corresponding datasets. The tasks we chose are listed in Table 5.

## D  Data Statistics

### D.1  Statistics of Instruction-Tuning Data

The statistics of the final instruction-tuning data for Domain-Curated-LM, Domain-Instruct-LM, Explore-LM, and Explore-LM-Ext in different domains are shown in Table 6.

### D.2  Statistics of Test Data

The statistics of our testbeds in different domains are shown in Table 7. Notably, the brainstorming and rewriting domains are characterized as open-ended, with no definitive standard answer available for evaluation within their testbed.

| Domains | Tasks |
|---|---|
| Brainstorming | task582_naturalquestion_answer_generation |
| | task669_ambigqa_answer_generation |
| | task898_freebase_qa_answer_generation |
| Rewriting | task045_miscellaneous_sentence_paraphrasing |
| | task132_dais_text_modification |
| | task177_para-nmt_paraphrasing |
| | task770_pawsx_english_text_modification |
| | task1614_sick_text_modify |
| | task1415_youtube_caption_corrections_grammar_correction |
| | task1557_jfleg_answer_generation |
| | task111_asset_sentence_simplification |
| | task933_wiki_auto_style_transfer |
| | task955_wiki_auto_style_transfer |
| | task927_yelp_negative_to_positive_style_transfer |
| | task928_yelp_positive_to_negative_style_transfer |
| Math | task085_unnatural_addsub_arithmetic |
| | task090_equation_learner_algebra |
| | task092_check_prime_classification |
| | task745_ai2_arithmetic_questions_arithmetic |
| | task751_svamp_subtraction_question_answering |
| | task752_svamp_multiplication_question_answering |
| | task753_svamp_addition_question_answering |
| | task754_svamp_common-division_question_answering |
| | task835_mathdataset_answer_generation |
| | task861_asdiv_addsub_question_answering |
| | task862_asdiv_multidiv_question_answering |
| | task863_asdiv_multiop_question_answering |
| | task864_asdiv_singleop_question_answering |
| | task865_mawps_addsub_question_answering |
| | task866_mawps_multidiv_question_answering |
| | task867_mawps_multiop_question_answering |
| | task868_mawps_singleop_question_answering |
| | task1726_mathqa_correct_answer_generation |

Table 5: Selected Domain-Specific Human-Curated datasets from SUPER-NATURALINSTRUCTION.

| Statistics | Brainstorming | Rewriting | Math |
|---|---|---|---|
| *Domain-Curated-LM* | | | |
| # Sampled training instances | 10,000 | 10,000 | 10,000 |
| Avg. instruction length | 50.57 | 74.10 | 65.55 |
| Avg. input length | 11.75 | 88.81 | 19.36 |
| Avg. output length | 2.42 | 90.50 | 1.14 |
| *Domain-Instruct-LM* | | | |
| # Maximum depth | 0 | 0 | 0 |
| # Sampled training instances | 10,000 | 10,000 | 10,000 |
| Avg. instruction length | 13.20 | 15.98 | 18.68 |
| Avg. input length | 24.21 | 17.86 | 3.14 |
| Avg. output length | 151.69 | 17.55 | 98.18 |
| *Explore-LM* | | | |
| # Maximum depth | 1 | 2 | 2 |
| # Sampled training instances | 10,000 | 10,000 | 10,000 |
| Avg. instruction length | 12.73 | 13.51 | 20.25 |
| Avg. input length | 20.12 | 22.48 | 10.57 |
| Avg. output length | 173.29 | 27.93 | 121.59 |
| *Explore-LM-Ext* | | | |
| # Maximum depth | 1 | 2 | 3 |
| # Sampled training instances | 16,000 | 32,000 | 64,000 |
| Avg. instruction length | 12.90 | 14.71 | 17.27 |
| Avg. input length | 21.57 | 20.24 | 12.88 |
| Avg. output length | 165.60 | 23.02 | 120.16 |

Table 6: Statistics of final instruction-tuning data for Domain-Curated-LM, Domain-Instruct-LM, Explore-LM, and Explore-LM-Ext in different domains.

| Statistics | Brainstorming | Rewriting | Math |
|---|---|---|---|
| # Questions | 208 | 94 | 500 |
| Avg. question length | 16.05 | 42.78 | 57.64 |
| Avg. answer length | - | - | 167.57 |

Table 7: Statistics of testbeds in different domains.

# E    Comparison with Methods for the General Domain

We conduct automatic comparison experiments with three well-known baselines for the general domain: Alpaca (Taori et al., 2023), Dromedary (Sun et al., 2023), and WizardLM (Xu et al., 2023a). In these experiments, Alpaca utilizes its entire dataset of 52,000 instruction-tuning instances, whereas Dromedary and WizardLM each draw a sample of 52,000 instances from their corresponding training sets. The experimental results for different domains are shown in Table 8, Table 9, and Table 10, respectively. The results show that Explore-LM outperforms the baselines across all domains.

| Automatic Comparison | Win:Tie:Lose | Beat Rate |
|---|---|---|
| Explore-LM vs Alpaca | 148:38:22 | 87.06 |
| Explore-LM-Ext vs Alpaca | 147:40:21 | **87.50** |
| Explore-LM vs Dromedary | 163:21:24 | 87.17 |
| Explore-LM-Ext vs Dromedary | 165:22:21 | **88.71** |
| Explore-LM vs WizardLM | 84:55:69 | 54.90 |
| Explore-LM-Ext vs WizardLM | 93:65:50 | **65.03** |

Table 8: Automatic comparison between Explore-LM and baselines in the brainstorming domain.

| Automatic Comparison | Win:Tie:Lose | Beat Rate |
|---|---|---|
| Explore-LM vs Alpaca | 22:52:20 | 52.38 |
| Explore-LM-Ext vs Alpaca | 25:55:14 | **64.10** |
| Explore-LM vs Dromedary | 40:40:14 | 74.07 |
| Explore-LM-Ext vs Dromedary | 43:42:9 | **82.69** |
| Explore-LM vs WizardLM | 23:49:22 | 51.11 |
| Explore-LM-Ext vs WizardLM | 20:60:14 | **58.82** |

Table 9: Automatic comparison between Explore-LM and baselines in the rewriting domain.

| Models | Accuracy Rate |
|---|---|
| Alpaca | 3.8 |
| Dromedary | 1.8 |
| WizardLM | 2.2 |
| Explore-LM | 6.8 |
| Explore-LM-Ext | **8.4** |

Table 10: Automatic comparison between Explore-LM and baselines in the math domain.

# F    Case Studies

In Table 11, Table 12, and Table 13, we show the case studies that examine the performance of Domain-Curated-LM, Domain-Instruct-LM, Explore-LM, Explore-LM-Ext, and ChatGPT in different domains respectively.

```
# Exploration Prompt

You are asked to propose some new sub-tasks for the target task given a list of existing sub-tasks
and another list of existing sibling tasks, then generate a set of examples for each new sub-task.
Each example consists of an instruction, an input, and an output.

Here are the requirements:
1. The skills required to perform a sub-task belong to the skills required to perform the target task
, and the former is a subset of the latter.
2. The skills required to perform a sibling task relate to the skills required to perform the target
task. There is an intersection of the former and the latter.
3. The sub-task and sibling task should focus on common domains, not specific domains.
4. A new sub-task is complementary to existing sub-tasks, and the addition of a new sub-task is
essential to the completion of the target task.
5. The new sub-task should be different from the existing sub-tasks and sibling tasks. The skills
required for a new sub-task should be designed to avoid overlapping with existing sub-tasks and
sibling tasks.
6. The instruction should be in English.
7. The instruction should be 1 to 2 sentences long. Either an imperative sentence or a question is
permitted.
8. The instruction should not contain specific examples and detailed content.
9. Try not to repeat the verb for each instruction in the examples to maximize diversity.
10. The instruction should be able to complete by a GPT language model. For example, the instruction
should not ask the assistant to create any visual or audio output. For another example, do not ask
the assistant to wake you up at 5 pm or set a reminder because it cannot perform any action.
11. You should create an appropriate input based on the instruction in an example, but the input
should not respond to the instruction. The input should involve realistic data and should not contain
 simple placeholders. The input should provide substantial content to make the instruction
challenging but do not exceed 200 words in general.
12. Note that some instructions do not require input. For example, when an instruction asks about
some general information of self-contained, eg: "What is the highest mountain in the world." or "
Please list 5 different fruits.", it is not necessary to provide a specific context. In this case, we
 simply put "<noinput>" in the input field.
13. You should generate an appropriate output according to the instruction and depending on the input
 in an example. Make sure the output is less than 200 words in general.
14. The response you generated should conform to the following format:
###
1. Instruction: ____
Input: ____
Output: ____
###
2. Instruction: ____
Input: ____
Output: ____
###

Target task: assistance_for_text_editting
Examples:
###
1. Instruction: Rewrite this text in another way
Input: People often think of cats as mysterious creatures because they are especially active at night
. Also, their eyes can glow an eerie green light in the dark.
Output: Cats are often considered mystical creatures because they are particularly active at night.
Also, their eyes can emit a strange green glow in dark environments.
###
2. Instruction:-Extend the content from the brief description below.
Input: This is a great phone with a high-resolution display.
Output: This is an excellent phone with a high-performance processor for faster performance and
smoother multitasking. Furthermore, it has a high-resolution display for crisp, vibrant images ,
allowing users to enjoy a better visual experience.
###

The list of already existing sub-tasks for this target task is: ['paraphrase', 'style_transfer', '
simplify_language'].
The list of already existing sibling tasks for this target task is: [].

The target task should be decomposed into a total of 8 diverse and complementary sub-tasks, and there
 are 3 existing sub-tasks. Generate 3 new sub-tasks with the corresponding reason, then list 10
examples of this new sub-task:
```

Figure 7: Prompt used for lookahead and backtracking explorations.

```
# Generation Prompt

You are asked to generate a set of examples for a new sub-task. Each example consists of an
instruction, an input, and an output.

Here are the requirements:
1. The skills required to perform a sub-task belong to the skills required to perform the target task
, and the former is a subset of the latter.
2. The instruction should be in English. The instruction should be 1 to 2 sentences long. Either an
imperative sentence or a question is permitted.
3. You should create an appropriate input based on the instruction in an example. The input should
involve realistic data and should not contain simple placeholders. The input should provide
substantial content to make the instruction challenging but do not exceed 200 words in general.
4. The input should include detailed content of a passage or an article if instructed, but not any
overview or description about it.
5. You should generate an appropriate output according to the instruction and depending on the input
in an example. Make sure the output is less than 200 words in general.
6. The response you generated should conform to the following format:
###
1. Instruction: ____
Input: ____
Output: ____
###
2. Instruction: ____
Input: ____
Output: ____
###

Target task: assistance_for_text_editting
Examples:
###
1. Instruction: Rewrite this text in another way
Input: People often think of cats as mysterious creatures because they are especially active at night
. Also, their eyes can glow an eerie green light in the dark.
Output: Cats are often considered mystical creatures because they are particularly active at night.
Also, their eyes can emit a strange green glow in dark environments.
###
2. Instruction:-Extend the content from the brief description below.
Input: This is a great phone with a high-resolution display.
Output: This is an excellent phone with a high-performance processor for faster performance and
smoother multitasking. Furthermore, it has a high-resolution display for crisp, vibrant images ,
allowing users to enjoy a better visual experience.
###

The list of already existing sub-tasks for this target task is: ['paraphrase', 'style_transfer', '
simplify_language'].
The list of already existing sibling tasks for this target task is: [].

List 10 examples of this new sub-task below:

New sub-task: paraphrase
Reason: The purposes of paraphrasing are to show understanding, make ideas accessible, translate
technical language, avoid plagiarism, compare perspectives, and aid memorization. The ability to
paraphrase skillfully is a valuable tool for write assistance.
```

Figure 8: Prompt used for domain-specific instruction-tuning data generation.

```
# Evaluation Prompt
You are a helpful and precise assistant for checking the quality of the answer.

[Question]
Please edit the following paragraph to add more details to make it more interesting.
I went to the park yesterday. I saw two puppies playing there. Then I went to the lake and saw a lot
of birds flying by. I ended up sitting on a bench and admiring the beauty of nature.

[Assistant 1]
Yesterday, I decided to take a stroll in the park. As I was walking, I noticed two adorable puppies
gleefully running around. Their playful barks and wagging tails were infectious, and I couldn't help
but smile. As I continued walking, I made my way over to the lake. There, I saw a flock of birds
soaring high above the water, creating an impressive sight. It was as if they were performing a dance
 in the sky. Mesmerized by their beauty, I found myself gravitating towards a bench, where I sat and
allowed myself to bask in the tranquility of nature. The sun was shining, the air was crisp, and
everything around me was peaceful. It was the perfect way to spend an afternoon.

[End of Assistant 1]

[Assistant 2]
Yesterday, I took a stroll in the park and witnessed two playful puppies running around. The sight of
 these adorable dogs brought joy to my day. After that, I ventured towards the lake and was amazed by
 the flock of birds flying in unison. I decided to sit on a bench and take in the tranquility of
nature.

[End of Assistant 2]

[System]
We would like to request your feedback on the performance of two AI assistants in response to the
user question displayed above.
Please evaluate the given four aspects: helpfulness, relevance, accuracy, level of details of their
responses.
Please first clarify how each response achieves each aspect respectively.
Then, provide a comparison on the overall performance between Assistant 1 and Assistant 2, and you
need to clarify which one is better than or equal to another. Avoid any potential bias and ensuring
that the order in which the responses were presented does not affect your judgment.
In the last line, order the two assistants. Please output a single line ordering Assistant 1 and
Assistant 2, where '>' means 'is better than' and '=' means 'is equal to'. The order should be
consistent to your comparison. If there is not comparison that one is better, it is assumed they have
 equivalent overall performance ('=').
```

Figure 9: Prompt used for automatic pair-wise evaluation.

| | Domain: Brainstorming |
|---|---|
| **Instruction:** Provides a solution to the following question: How to maximize your productivity? | |
| **Domain-Curated-LM** | Prioritize your tasks |
| **Domain-Instruct-LM** | Here is a 5-step guide to maximize your productivity:
1. Prioritize - Make a realistic to-do list and rank each task based on its importance and deadline.
2. Plan - Spend time planning out your workday to ensure you make the most of your time and avoid distractions.
3. Take Breaks - Break up your work with regular short breaks to refresh your mind and energize yourself.
4. Focus - Reduce distractions by turning off notifications and putting your phone away.
5. Communicate - Set clear expectations with your team or colleagues to avoid confusion and duplication of efforts. |
| **Explore-LM** | Sure, I'd be happy to help you with that. Here are some tips to maximize your productivity:
1. Set clear goals: Set clear and specific goals for what you would like to achieve. This will help you stay focused and motivated.
2. Break down tasks: Break down larger tasks into smaller, more manageable ones. This will make them less overwhelming and easier to accomplish.
3. Use a to-do list: Make a to-do list for each day, and prioritize tasks based on their importance and urgency.
4. Schedule your time: Schedule specific times to work on important tasks. This will help you stay focused and avoid distractions.
5. Take breaks: Take regular breaks to avoid burnout and maintain focus. Try to take breaks away from your workspace to help you recharge.
6. Use a timer: Use a timer to help you stay focused and work for specific intervals of time. This can help you stay on track and avoid procrastination.
7. Minimize distractions: Minimize distractions such as social media, email notifications, and phone calls. Instead, use tools such as Freedom or Cold Turkey to block distractions.
8. Prioritize self-care: Prioritize self-care activities such as exercise, meditation, or spending time with loved ones. This will help you maintain your mental and physical health, which is essential for productivity.
9. Outsource tasks: Outsource tasks that you are not good at or don't enjoy doing. This will help you focus on your strengths and improve your efficiency.
10. Automate tasks: Automate repetitive tasks such as bookkeeping, scheduling, or email responses. This will save you time and increase your productivity in the long run. |

| | |
|---|---|
| **Explore-LM-Ext** | Sure, let's brainstorm some ways to maximize your productivity:
1. Prioritize your tasks: Make a list of everything you need to do, and prioritize your tasks based on their importance and deadline.
2. Use time management techniques: Use techniques such as the Pomodoro Technique or time blocking to manage your time effectively and reduce distractions.
3. Take breaks: Taking regular breaks can help you recharge and stay focused for longer.
4. Automate repetitive tasks: Automate tasks such as email responses or social media updates to save time and reduce mental clutter.
5. Delegate tasks: Delegate tasks to others when possible, to free up time for more important tasks.
6. Manage your energy: Take steps to manage your energy throughout the day, such as getting enough sleep, eating well, and exercising regularly.
7. Eliminate distractions: Eliminate distractions such as social media or email notifications, and work in a quiet environment where possible.
8. Get organized: Keep your workspace organized and decluttered to reduce mental clutter and improve focus.
9. Learn to say no: Learn to say no to tasks and commitments that are not a priority or will take up too much time.
10. Celebrate progress: Celebrate your progress and achievements regularly to stay motivated and encourage continued productivity. |
| **ChatGPT** | 1. Set clear goals and prioritize tasks: Determine what tasks are most important and set achievable goals to help you stay focused and motivated.
2. Use time-blocking techniques: Schedule blocks of time for specific tasks, and avoid distractions during those periods to maximize productivity.
3. Take breaks and manage your energy: Taking regular breaks can help you recharge and stay focused. Consider incorporating physical activity or mindfulness practices into your breaks to help manage stress and improve concentration.
4. Use productivity tools: There are a variety of apps and tools available to help you stay organized, manage your time, and automate repetitive tasks.
5. Delegate tasks: Identify tasks that can be delegated to others, freeing up time for you to focus on more important responsibilities.
6. Practice good time management habits: Avoid procrastination, limit multitasking, and establish a routine that works for you.
7. Stay motivated: Find ways to stay motivated and inspired, such as setting rewards for completing goals or seeking support from colleagues or mentors. |

Table 11: A comparison case in the brainstorming domain.

| Domain: Rewriting | |
|---|---|
| **Instruction:** Edit the following article to add more details and descriptions to bring it to life. I went to a new restaurant yesterday and ordered a steak and a glass of red wine. The food tastes good, but the waiters have a bad attitude. | |
| **Domain-Curated-LM** | The food tastes good, but the waiters have a bad attitude. |
| **Domain-Instruct-LM** | Yesterday, I went to a new restaurant and ordered a steak and a glass of red wine. The food tasted delicious, but the waiters had a bad attitude. |
| **Explore-LM** | Yesterday, I had the pleasure of dining at a new restaurant that serves delectable steaks and an exquisite selection of wines. The perfectly seasoned filet mignon melted in my mouth, while the pairing of the robust red wine complemented the flavors of the meat. However, I was disappointed in the attitude of the waiters, who seemed to be more interested in their cell phones than in providing excellent service. |
| **Explore-LM-Ext** | Yesterday, I visited a trendy steakhouse with high hopes. When my waiter brought the menu, I noticed that it was well-stocked with prime cuts of meat and an extensive list of wine pairings. I decided to order the filet mignon and a glass of red wine. Actually, the food was delicious, but the waiters seemed to have a chip on their shoulder, acting as if they were doing us a favor by serving us. Overall, the experience was mixed, and I'm still on the fence about returning. |
| **ChatGPT** | Yesterday, I decided to try out a new restaurant that I had heard about. Upon arriving, I was immediately impressed with the chic and modern decor. I took a seat and was greeted by a waiter who seemed uninterested in helping me. Despite this, I ordered a perfectly cooked steak and a glass of rich, full-bodied red wine. 

 The steak was cooked to absolute perfection, with a crispy seared exterior and a juicy, tender center. The first bite was a burst of flavor that left me craving more. The red wine perfectly complemented the steak, with its robust flavor and notes of blackberry and oak. 

 However, my dining experience was marred by the poor attitude of the waiters. They seemed uninterested in serving me and were slow to respond to my requests. This put a damper on an otherwise enjoyable meal. 

 Despite the rude service, I would still recommend this restaurant for the quality of the food alone. It's a shame that the waitstaff didn't match the caliber of the cuisine. |

Table 12: A comparison case in the rewriting domain.

| | **Domain: Math** |
|---|---|
| **Instruction:** What is the maximum number of square inches in the area of a rectangle with a perimeter of 12 inches? | |
| **Domain-Curated-LM** | 159 |
| **Domain-Instruct-LM** | Explanation: Let $l$ and $w$ be the length and width of the rectangle, respectively. We know that $2l + 2w = 12$, so $l + w = 6$. The area of the rectangle is $A = lw$. We want to maximize $A$ subject to the constraint $2l + 2w = 12$. Using the method of Lagrange multipliers, we get $l = 3, w = 4$, and $A = 12$. Therefore, the maximum number of square inches in the area of the rectangle is $\boxed{12}$. Answer: $\boxed{12}$. |
| **Explore-LM** | Explanation: Let $l$ and $w$ be the length and width of the rectangle, respectively. We know that $2l + 2w = 12$, so $l + w = 6$. The area of the rectangle is $A = lw$. We want to maximize $A$ subject to the constraint $l + w = 6$. Using the method of Lagrange multipliers, we get the system of equations $l = w, l + w = 6$, and $\nabla A = \lambda \nabla (l + w - 6)$. Solving this system of equations, we get $l = w = 3$. Therefore, the maximum number of square inches in the area of the rectangle is $3^2 = 9$. Answer: $\boxed{9}$. |
| **Explore-LM-Ext** | Explanation: Let $l$ and $w$ be the length and width of the rectangle, respectively. Since the perimeter is 12 inches, we have $2l + 2w = 12$, or $l + w = 6$. We want to maximize the area, which is given by $A = lw$. From the equation for the perimeter, we have $l = 6 - w$. Substituting this into the equation for the area, we get $A = w(6 - w) = 6w - w^2$. To maximize this expression, we take the derivative with respect to $w$ and set it equal to 0: $\frac{d}{dw}(6w - w^2) = 6 - 2w = 0$. Solving for $w$, we get $w = 3$, which implies that $l = 3$. Therefore, the maximum area is $A = 3(3) = 9$. Answer: $\boxed{9}$ square inches. |
| **ChatGPT** | Explanation: Let the width of the rectangle be $w$ and the length be $l$. The perimeter of the rectangle is given by $2(l + w) = 12$, which simplifies to $l + w = 6$. We know that the area of a rectangle is given by $A = lw$. We want to maximize the area of the rectangle, so we need to maximize $A$ subject to the constraint $l + w = 6$. Using the method of Lagrange multipliers, we set up the following equations: $$\nabla A = \lambda \nabla (l + w)$$ $$lw = \lambda(1, 1)$$ $$l + w = 6$$ Taking partial derivatives, we get: $$\frac{\partial A}{\partial l} = w$$ $$\frac{\partial A}{\partial w} = l$$ And we have: $$\frac{w}{\lambda} = 1$$ $$\frac{l}{\lambda} = 1$$ Multiplying both equations by $\lambda$ and setting them equal to each other, we get $w = l$. Substituting this into the constraint equation, we get $2l = 6$, so $l = 3$ and $w = 3$. Therefore, the maximum area of the rectangle is $A = lw = 3 \cdot 3 = \boxed{9}$ square inches. Answer: $\boxed{9}$ square inches. |

| | |
|---|---|
| **Standard Answer** | Since the perimeter is 12, the sides of the rectangle add up to $12/2 = 6$. Let $x$ be one side length of the rectangle. Then the other side length is $6 - x$, so the area is $$x(6 - x) = 6x - x^2.$$ Completing the square, we get $$-x^2 + 6x = -x^2 + 6x - 9 + 9 = 9 - (x - 3)^2.$$ Thus, the maximum area of the rectangle is $\boxed{9}$ square inches, which occurs for a $3 \times 3$ square. |

Table 13: A comparison case in the math domain.