# OpenReview forum: "Explore-Instruct: Enhancing Domain-Specific Instruction Coverage through Active Exploration"
_EMNLP/2023/Conference — EMNLP 2023 Main_

### Official Review · Reviewer_TUPc · 2023-07-28

**Soundness:** 4

**Excitement:**

4: Strong: This paper deepens the understanding of some phenomenon or lowers the barriers to an existing research direction.

**Paper Topic And Main Contributions:**

This paper introduces Explore-Instruct, an instruction-dataset-generation approach that models domains / instructions as a graph, and generates data by traversing domains in a DFS-style manner, exploring until a max depth and/or max breadth parameter is reached. Alpaca-style generation is used for data generation while in a domain, while llama-7b models are trained. Model-based and human-based evaluation suggests that the approach performs better than domain-specific models, although it lags somewhat behind ChatGPT.


**Questions For The Authors:**

A. How does your approach compare to baseline alpaca [2], self-instruct [3], or longform [4]? Do you have experiments for comparison?

B. How does your approach compare to evol-instruct [1]?

C. It's interesting to me that better domain coverage would aid a dataset like MATH, which is very domain specific. Why do you think explore-instruct improves here?

D. Does domain-aware self-instruct use domain-specific seed examples for e.g. generating data for the math domain and evaluation? This detail wasn't entirely clear to me in the description in 4.1.

[1] https://arxiv.org/abs/2304.12244
[2] https://crfm.stanford.edu/2023/03/13/alpaca.html
[3] https://arxiv.org/abs/2212.10560
[4] https://arxiv.org/abs/2304.08460

**Reasons To Accept:**

Straightforward method explored well, with good results and a few interesting analyses around the effect of data quality. The approach seems to work well as a way to increase the diversity of an instruction set, keeping the overlap between instructions relatively low. Performing a graph search over the space of domains is an interesting and novel idea, and seems to perform reasonably well.

**Reasons To Reject:**

Lack of other baselines - the authors note there is much similar work, but don’t compare to these alternatives. For example, it would be good to compare to baseline self-instruct, or alpaca, to show the improvement of using this method against these other ones (beyond the domain-aware self-instruct, which to my understanding would be similar to alpaca). Such comparisons would further highlight the improvements of the method - althought I think the current experiments show fairly well that domain-level exploration is useful. Although it is concurrent work, I would be especially interested in comparisons to Evol-Instruct (the approach behind WizardLM), which also takes a graph search-style approach to generating instruction data (but on a sample level, rather than a domain level).

Overall, I think this is solid and interesting work, and lean accept.


**Reproducibility:**

4: Could mostly reproduce the results, but there may be some variation because of sample variance or minor variations in their interpretation of the protocol or method.

**Reviewer Confidence:**

4: Quite sure. I tried to check the important points carefully. It's unlikely, though conceivable, that I missed something that should affect my ratings.

---

> ### Author Rebuttal · Authors · 2023-08-28
>
> We sincerely appreciate the time and effort you put into reviewing our paper. We appreciate your insightful feedback and we are thankful for your recognition of our work's novelty and significance. In the subsequent points, we will carefully address each of your concerns.
>
> **Q1: Regarding the comparison with other baseline datasets.**
>
> **A1:** We conducted automatic comparison experiments with three well-known baseline models: **Alpaca**, **WizardLM**, and **Dromedary**. In these experiments, Alpaca utilized its entire dataset of 52,000 instruction-tuning instances, whereas WizardLM and Dromedary each drew a sample of 52,000 instances from their corresponding training sets. The experimental results are shown in the tables below, and **our Explore-LM outperforms the baseline models across all domains**. We will include these additional results in the revised version.
>
> | Automatic Comparison | Win:Tie:Lose | Beat Rate |
> | :-: | :-: | :-: |
> | Explore-LM vs Alpaca | 148:38:22 | 87.06 |
> | Explore-LM-Ext vs Alpaca | 147:40:21 | 87.50 |
> | Explore-LM vs WizardLM | 84:55:69 | 54.90 |
> | Explore-LM-Ext vs WizardLM | 93:65:50 | 65.03 |
> | Explore-LM vs Dromedary | 163:21:24 | 87.17 |
> | Explore-LM-Ext vs Dromedary | 165:22:21 | 88.71 |
>
> **Table 1: Evaluation results in the brainstorming domain.**
>
> | Automatic Comparison | Win:Tie:Lose | Beat Rate |
> | :-: | :-: | :-: |
> | Explore-LM vs Alpaca | 22:52:20 | 52.38 |
> | Explore-LM-Ext vs Alpaca | 25:55:14 | 64.10 |
> | Explore-LM vs WizardLM | 23:49:22 | 51.11 |
> | Explore-LM-Ext vs WizardLM | 20:60:14 | 58.82 |
> | Explore-LM vs Dromedary | 40:40:14 | 74.07 |
> | Explore-LM-Ext vs Dromedary | 43:42:9 | 82.69 |
>
> **Table 2: Evaluation results in the rewriting domain.**
>
> | Models | Accuracy Rate |
> | :-: | :-: |
> | Alpaca | 3.8 |
> | WizardLM | 2.2 |
> | Dromedary | 1.8 |
> | Explore-LM | 6.8 |
> | Explore-LM-Ext | 8.4 |
>
> **Table 3: Evaluation results in the math domain.**
>
> **Q2: Regarding the comparison with Evol-Instruct.**
>
> **A2:** Evol-Instruct includes two distinct operations, in-depth evolution, which emphasizes the generation of progressively complex instructions rather than augmenting their diversity, and in-breadth evolution, which has the potential to extend across various domains, such as transitioning from mathematics to physics (e.g., Figure 1 in WizardLM [1]). Consequently, Evol-Instruct tends to be inadequate for enhancing the coverage of instruction-tuning data within a specific domain.
>
> [1] Xu C, Sun Q, Zheng K, et al. Wizardlm: Empowering large language models to follow complex instructions[J]. arXiv preprint arXiv:2304.12244, 2023.
>
> **Q3: Regarding the better domain coverage for MATH dataset.**
>
> **A3:** The MATH dataset includes a diverse range of mathematical problems with varying degrees of difficulty and categories, such as algebra, geometry, and calculus. Thus, solving questions from different categories requires not identical but complementary skills and knowledge. This is what the proposed Explore-Instruct aims to achieve by promoting task diversities and maintaining connections simultaneously.
>
> **Q4: Regarding the details about seed examples in Domain-Aware Self-Instruct.**
>
> **A4:** In Domain-Aware Self-Instruct, an initial set of domain-specific seed collections with diverse instructions is employed. During each iteration, two demonstrations are randomly selected from the seed collection. Subsequently, the newly generated instruction data is integrated into the seed collection, and the process is reiterated. We will include the above clarifications in the revised version.
>
> We sincerely thank the reviewer again for your valuable suggestions! Please let us know if you have any further questions, and we are happy to discuss further.

---

### Official Review · Reviewer_Rfvu · 2023-08-04

**Soundness:** 4

**Excitement:**

3: Ambivalent: It has merits (e.g., it reports state-of-the-art results, the idea is nice), but there are key weaknesses (e.g., it describes incremental work), and it can significantly benefit from another round of revision. However, I won't object to accepting it if my co-reviewers champion it.

**Missing References:**

LaMini-LM: A Diverse Herd of Distilled Models from Large-Scale Instructions.  Wu et al., 2023

Dynosaur: A Dynamic Growth Paradigm for Instruction-Tuning Data Curation. Yin et al., 2023


**Paper Topic And Main Contributions:**

Current instruction-tuning data generation methods often lack sufficient representation of specific domains, limiting nuanced understanding and interactions within those areas. To address this gap, the authors introduce a new method called Explore-Instruct. This approach actively enhances data coverage for domain-specific instruction-tuning by lookahead and backtrack exploration with LLMs. The authors’ method outperforms several baselines in terms of diversity and quality, even those using domain-specific data enhancement techniques. Also, it brings better downstream performance on specific domains like maths, brainstorming and rewriting.

**Questions For The Authors:**

1.	What is the prompt used for the Self-Instruct baselines?
2.	What is the cost of the entire method?
3.	Is there any experiment on the GSM8K dataset? This might be a more common one that people test model’s math reasoning ability?


**Reasons To Accept:**

1.	Great motivation and interesting research question: Domain-specific instruction generation is very important because it could enable everyone to design personalized instruction-tuned models and make instruction-tuned models satisfy the need of experts. Most current instruction generation methods don’t consider this point and generate sth very general. The authors design two types of exploration strategies: lookahead and backtrack exploration, that greatly contribute to the depth and breadth of the trained tasks. The method is reasonable to me.
2.	Comprehensive investigation on the quality of generated instruction data: I'm glad to see that the authors utilize different metrics to evaluate the data quality and diversity. The comprehensive evaluation helps us better understand the effect of proposed methods. I’m especially impressed by the win-lose comparison with domain curated and instruction tuned LMs and the tendency when increasing the number of training instances.


**Reasons To Reject:**

1.	A bit unfair comparison with Domain-aware Self-instruct: As the authors mentioned in Line 281, Self-instruct doesn’t take any exploration steps. Therefore, in terms of cost, building Explore-Instruct might be much more expensive than the simple Self-instruct. Also, more details about the reproduced Self-instruct is needed, e.g., prompt contents, steps of controlling diversity, etc. The seed task examples in Self-instruct prompt might be vital for the performance. If the examples are more domain-specific and you can provide a diverse set of them, the generated instruction may be diverse as well.
2.	Lack of more comprehensive test domains: I can only see brainstorming, maths and rewriting in the evaluation parts. However, when people talk about “domain-specific”, it is more natural to think of science domains like biology and medical domains. Rewriting is a general domain that may not need too much expert knowledge. If the paper really claims that the method can help generate domain-specific instructions, it is better to test more scenarios.
3.	Lack of fair comparison with other datasets like WizardLM and Dromedary: These two datasets also use some similar methods to further extend the existing instructions. But there’s no direct comparison with these two datasets and the methods they use.


**Reproducibility:**

3: Could reproduce the results with some difficulty. The settings of parameters are underspecified or subjectively determined; the training/evaluation data are not widely available.

**Reviewer Confidence:**

4: Quite sure. I tried to check the important points carefully. It's unlikely, though conceivable, that I missed something that should affect my ratings.

---

> ### Author Rebuttal · Authors · 2023-08-28
>
> We sincerely appreciate the time and effort you put into reviewing our paper. We appreciate your insightful feedback and are pleased that you find our work to be both motivating and interesting. We will carefully address each of your concerns in the following points.
>
> **Q1: Regarding the cost comparison between Domain-Aware Self-Instruct and Explore-Instruct.**
>
> **A1:** In contrast to the Domain-Aware Self-Instruct approach, the primary additional cost associated with the Explore-Instruct method comes mainly from the exploration process. Since this exploration occurs at the task level and allows for simultaneous exploration of multiple tasks, the associated costs are relatively minimal when compared to the generation of instruction-tuning data. To illustrate, consider the creation of a dataset for the rewriting domain consisting of 10,000 instances, which, without taking repetitive filtering into account, requires approximately **1,000 API calls * (400 prompt tokens + 600 output tokens) = 1,000,000 tokens** when generating 10 items concurrently. Conversely, the exploration of a hierarchical structure with a maximum depth of 2 and 10 subtasks at each level requires a mere **11 API calls * (700 prompt tokens + 200 outputs tokens) = 9,900 tokens** when exploring 10 subtasks simultaneously. As a result, the additional cost incurred is **less than 1%**. We will include the above clarifications in the revised version.
>
> **Q2: Regarding the details about Domain-Aware Sefl-Instruct.**
>
> **A2:** In the Domain-Aware Self-Instruct approach, we try our best to ensure the diversity of the instruction data generation process. The following components describe the prompt, seed task examples, and diversity filter in detail.
>
> **Prompt:** The prompt employed by Domain-Aware Self-Instruct in generating instruction-tuning data is similar to that of Explore-Instruct. Within the generation prompt, an explicit emphasis is placed on generating a diverse range of instructions to minimize redundancy between the new instructions and those already existing.
>
> **Seed Task Examples:** In Domain-Aware Self-Instruct, an initial set of domain-specific seed examples with diverse instructions is employed. During each iteration, two demonstrations are randomly selected from the seed collection. Subsequently, the newly generated instruction data is integrated into the seed collection, and the process is reiterated.
>
> **Diversity Filter:** The diversity filter applied during data generation is based on the ROUGE-L overlap between the instructions. Specifically, a generated instruction is preserved only if its ROUGE-L overlap with any pre-existing instructions falls below a given threshold.
>
> We will include the above clarifications in the revised version.
>
> **Q3: Regarding more comprehensive test domains.**
>
> **A3:** The science domains like biology and medical domains are orthogonal to the definition of domain employed in this study. Science domains represent broader fields of knowledge, while specific use case domains (discussed in this paper) represent specialized areas of application or functionality within those fields. Besides the domains discussed in our paper (namely brainstorming, rewriting, and math), other similar examples include assistance in data analysis, collaboration in programming, and engagement in interactive role-playing. Therefore, utilizing science domains as experimental grounds is not included in the focus of our research.
>
> **Q4: Regarding the comparison with other baseline datasets.**
>
> **A4:** We conducted automatic comparison experiments with three well-known baseline models: **Alpaca**, **WizardLM**, and **Dromedary**. In these experiments, Alpaca utilized its entire dataset of 52,000 instruction-tuning instances, whereas WizardLM and Dromedary each drew a sample of 52,000 instances from their corresponding training sets. The experimental results are shown in the tables below, and **our Explore-LM outperforms the baseline models across all domains**. We will include these additional results in the revised version.
>
> | Automatic Comparison | Win:Tie:Lose | Beat Rate |
> | :-: | :-: | :-: |
> | Explore-LM vs Alpaca | 148:38:22 | 87.06 |
> | Explore-LM-Ext vs Alpaca | 147:40:21 | 87.50 |
> | Explore-LM vs WizardLM | 84:55:69 | 54.90 |
> | Explore-LM-Ext vs WizardLM | 93:65:50 | 65.03 |
> | Explore-LM vs Dromedary | 163:21:24 | 87.17 |
> | Explore-LM-Ext vs Dromedary | 165:22:21 | 88.71 |
>
> **Table 1: Evaluation results in the brainstorming domain.**
>
> | Automatic Comparison | Win:Tie:Lose | Beat Rate |
> | :-: | :-: | :-: |
> | Explore-LM vs Alpaca | 22:52:20 | 52.38 |
> | Explore-LM-Ext vs Alpaca | 25:55:14 | 64.10 |
> | Explore-LM vs WizardLM | 23:49:22 | 51.11 |
> | Explore-LM-Ext vs WizardLM | 20:60:14 | 58.82 |
> | Explore-LM vs Dromedary | 40:40:14 | 74.07 |
> | Explore-LM-Ext vs Dromedary | 43:42:9 | 82.69 |
>
> **Table 2: Evaluation results in the rewriting domain.**
>
> | Models | Accuracy Rate |
> | :-: | :-: |
> | Alpaca | 3.8 |
> | WizardLM | 2.2 |
> | Dromedary | 1.8 |
> | Explore-LM | 6.8 |
> | Explore-LM-Ext | 8.4 |
>
> **Table 3: Evaluation results in the math domain.**
>
> **Q5: Regarding the experiments on GSM8K dataset.**
>
> **A5:** In our study, we employ the MATH dataset for evaluation in the math domain. This is because it includes a diverse range of mathematical problems with varying degrees of difficulty and categories, such as algebra, geometry, and calculus. This choice aligns with the research problem of this paper: domain-specific performances. In contrast, the GSM8K dataset primarily consists of grade school mathematical word problems. These can be more accurately characterized as a specific task rather than a comprehensive domain, akin to the paraphrasing task within the rewriting domain.
>
> **Q6: Regarding the missing references.**
>
> **A6:** We will add these references in the revised version accordingly.
>
> We sincerely thank the reviewer again for your valuable suggestions. If you have any further questions, please let us know and we will be happy to discuss them with you further.

---

### Official Review · Reviewer_Jzc1 · 2023-08-05

**Soundness:** 4

**Excitement:**

4: Strong: This paper deepens the understanding of some phenomenon or lowers the barriers to an existing research direction.

**Paper Topic And Main Contributions:**

This work presents an approach to enhance the data coverage to be used in domain-specific instruction-tuning through active exploration with LLMs.

**Questions For The Authors:**

I think this work might benefit from human assessment of the in-domain instructions in addition to model evaluation. Could you discuss the implications of this further?

**Reasons To Accept:**

Current methodologies for domain-specific instruction-tuning data often sparsely cover the full range of a domain. This problem is therefore relevant and highly important to address for domain-specific applications. The authors' approach is novel and well-argued.


**Reasons To Reject:**

The work does not discuss the generation of more complex, multi-step reasoning instructions, nor about how to mitigate instruction bias that might lead to the generation of incorrect or harmful text.

**Reproducibility:**

4: Could mostly reproduce the results, but there may be some variation because of sample variance or minor variations in their interpretation of the protocol or method.

**Reviewer Confidence:**

3: Pretty sure, but there's a chance I missed something. Although I have a good feel for this area in general, I did not carefully check the paper's details, e.g., the math, experimental design, or novelty.

---

> ### Author Rebuttal · Authors · 2023-08-28
>
> We sincerely appreciate your valuable feedback and are grateful that you find our approach to be novel and well-argued.
>
> **Q1: Regarding the generation of more complex reasoning instructions.**
>
> **A1:** The motivation of Explore-Instruct is to augment domain-specific instructional coverage and subsequently improve the performance of the fine-tuned model. It is essential to note that, in some domains, increasing the instruction complexity does not necessarily guarantee enhanced domain-specific performance. Nevertheless, in the case of the MATH dataset, a positive correlation exists between the complexity of instructions and the overall performance. This demonstrates that our approach effectively promotes reasoning capabilities.
>
> **Q2: Regarding the instruction bias problem.**
>
> **A2:** Our approach is orthogonal to the works aimed at diminishing instruction bias, signifying that the strategies for reducing such bias can typically be incorporated into our method. For example, the utilization of reinforcement learning from human feedback in ChatGPT has the potential to alleviate the production of inaccurate or detrimental text within the context of our proposed framework.
>
> **Q3: Regarding the human assessment of the instructions.**
>
> **A3:** In the approach we proposed, assessing the generated instruction-tuning data requires the analysis of two critical factors. The first factor is the data quality, which can be referred to in previous works where each instruction, input, and output triplet undergoes human evaluation to determine its accuracy and effectiveness in accomplishing the task. The second crucial factor involves the data's diversity, which is non-trivial to estimate by human evaluators. We will supplement the human evaluation results of the data quality in the revised version.
>
> Thanks again for your review! If you have any additional questions, please let us know and we would be happy to discuss with you further.

---

### Meta-Review · Area_Chair_BKiy · 2023-09-19

**Recommendation:** 5

**Metareview:**

In this paper the authors propose an approach to enhance the data coverage to be used in domain-specific instruction-tuning through active exploration via Large Language Models (LLMs). The proposed method outperforms several baselines in terms of diversity and quality, even those using domain-specific data enhancement techniques. Also, it brings better downstream performance on specific domains like maths, brainstorming and rewriting.

Pros:

It is a well-motivated, highly timely and relevant research topic

Comprehensive investigation on the quality of generated instruction data

The approach seems to work well at  increasing the diversity of an instruction set, keeping the overlap between instructions relatively low.

The authors also performed a graph search over the space of domains which is an interesting and novel idea, and seems to perform reasonably well.

Cons:

Needs more comprehensive test domains, which are more “domain-intensive” like biology or medical (than maths, brainstorming and rewriting)

Needs more fair comparison with other baselines.

The original work will also benefit from incorporating the additional discussion points/experiments conducted during rebuttal.

---

### Decision · Program_Chairs · 2023-10-07

**Decision:**

Accept-Main

**Comment:**

In this paper the authors propose an approach to enhance the data coverage to be used in domain-specific instruction-tuning through active exploration via Large Language Models (LLMs). The proposed method outperforms several baselines in terms of diversity and quality, even those using domain-specific data enhancement techniques. Also, it brings better downstream performance on specific domains like maths, brainstorming and rewriting.

Pros:

It is a well-motivated, highly timely and relevant research topic

Comprehensive investigation on the quality of generated instruction data

The approach seems to work well at  increasing the diversity of an instruction set, keeping the overlap between instructions relatively low.

The authors also performed a graph search over the space of domains which is an interesting and novel idea, and seems to perform reasonably well.

Cons:

Needs more comprehensive test domains, which are more “domain-intensive” like biology or medical (than maths, brainstorming and rewriting)

Needs more fair comparison with other baselines.

The original work will also benefit from incorporating the additional discussion points/experiments conducted during rebuttal.